# Update on Biomarkers Associated with Large-Artery Atherosclerosis Stroke

**DOI:** 10.3390/biom13081251

**Published:** 2023-08-16

**Authors:** Madalena Rosário, Ana Catarina Fonseca

**Affiliations:** 1Stroke Unit, Neurology, Neuroscience Department, Hospital de Santa Maria, Centro Hospitalar Universitário Lisboa Norte, 1649-028 Lisboa, Portugal; 2Centro de Estudos Egas Moniz, Faculdade de Medicina da Universidade de Lisboa, 1649-028 Lisboa, Portugal; 3Instituto de Medicina Molecular João Lobo Antunes, 1649-028 Lisboa, Portugal

**Keywords:** atherosclerosis, stroke, carotid, stenosis, biomarker, plaque, inflammation, IL-6, metalloproteinases, lipids

## Abstract

Intracranial and extracranial large-artery atherosclerosis (LAA) are a main cause of ischemic stroke. Biomarkers may aid in the diagnosis of LAA and help to stratify patients’ risk of stroke. We performed a narrative review of the literature, mainly published in the last five years, with the aim of identifying biomarkers associated either with intracranial or extracranial LAA in humans. Several potential biomarkers of LAA, mainly related to lipidic pathways and inflammation, have been studied. Diagnostic biomarkers of LAA were evaluated by measuring biomarkers levels in patients with LAA stroke and other stroke etiologies. Some biomarkers were associated with the functional prognosis of LAA stroke patients. Increased levels of IL-6 and sLOX-1 were associated with a risk of progression of carotid atherosclerotic disease. Findings support the notion that the immune system plays a central role in the pathogenesis of LAA. Overall, in most studies, results were not externally validated. In the future, biomarkers could be useful for the selection of patients for clinical trials. To adopt these biomarkers in clinical practice, we will need robust multicentric studies proving their reproducibility and a clear practical applicability for their use.

## 1. Introduction

Large-artery atherosclerosis (LAA) is one of the most common causes of ischemic stroke (IS). Its definition implies clinical and brain imaging findings of either significant stenosis or occlusion of a major brain artery (carotid or vertebral arteries) or of a branch of a cortical artery (anterior, middle or posterior cerebral arteries) [1]. The other etiological subtypes of IS according to the TOAST classification are cardioembolism, small vessels occlusion (SVO), stroke of other etiology (e.g., dissections, vasculitis) and stroke of undetermined etiology [2].

LAA can cause a stroke by two possible pathophysiological mechanisms: by hypoperfusion due to a hemodynamically stenotic vessel, or by atheroembolism when there is plaque rupture or ulceration with thrombus formation and an upstream embolism [3].

As the name implies, atherosclerosis is the underlying pathophysiology in an LAA stroke. Atheroma is typically located at the bifurcation of arteries, where turbulent flow is highest. Several mechanisms contribute to the formation and development of atheromatous lesions. Initially, induced endothelial dysfunction, potentiated by inflammation and hypercholesterolemia, leads to increased permeability, with entrance of oxidated low-density lipoproteins in the subendothelial space of the intima. Meanwhile, at the endothelial surface, the expression of adhesion molecules initiates platelet aggregation and lymphocyte/monocyte adhesion and infiltration. In the intima, these monocytes mature into macrophages, which take up oxidized low-density lipoprotein, and transform into foam cells. As this occurs, vascular smooth muscle cells shift from a contractile phenotype to an active synthetic phenotype, producing extracellular matrix and gradually transforming the lesion into a fibrous plaque [3]. Unstable plaques usually contain macrophages and T lymphocytes that secrete cytokines [4]. During this process, drivers of atheroma formation that are being produced can be released and detected in the bloodstream. These substances can be potentially useful biomarkers to clarify stroke etiology and to stratify patients regarding their risk of progression of the plaque of atheroma or stroke.

In this article, we aimed to perform a narrative review of the literature published in the last five years regarding biomarkers associated with LAA stroke in humans.

## 2. Biomarkers and Their Usefulness in Stroke

Biomarkers are objective indicators used to assess normal or pathological processes, evaluate responses to medical interventions and/or predict outcomes [5]. They can be biochemically measurable components, genetic information or physical characteristics of tissue captured by imaging techniques. The ideal biomarker should be easily accessible, standardized, highly sensitivity and specific, easily interpretable, cost-effective and have added value.

Biomarkers can help in stroke diagnosis, particularly in LAA stroke, for particularly two reasons:-Firstly, even though stenosis severity is the primarily used parameter for risk evaluation, being recommended by medical guidelines to decide treatment (medical vs. surgical), it is known that several high-grade stenoses are asymptomatic, with a low rate of symptomatic conversion. On the contrary, there appears to be a higher risk in low-grade stenosis, with a substantial proportion of symptomatic patients and a risk of recurrent ipsilateral stroke of up to 8% at three years [6]. Biomarkers can help to understand the real risk of symptomatic conversion of these patients and allow clinicians to act accordingly.-Secondly, LAA biomarkers may be useful to identify the stroke cause in patients classified as having an undetermined stroke etiology. The problem persists when patients have more than one possible cause for a stroke [5,7]. Having a biomarker that provides clues to a possible etiology may allow the clinician to better tailor preventive measures to the individual patient and plan their follow-up.

## 3. Biomarkers Associated with LAA Stroke

Several studies have identified different substances which can be biomarkers of LAA stroke.

### 3.1. Chemical Biomarkers

Cholesterol particles remain one of the most recognized biomarkers of LAA stroke. Low-density lipoprotein cholesterol (LDL-c) is one of the most well-known biomarkers associated with LAA. It is independently associated with the presence and extent of subclinical early systematic atherosclerosis. It is currently used as a target for medical treatment [8,9].

More recently, some studies analyzed other lipid parameters, which may be helpful in predicting vascular risk.

A low HDL-C level was shown to be strongly associated with the presence of symptomatic intracranial atherosclerotic stenosis [10]. There was an inverse relationship between the level of HDL-C and the presence of symptomatic intracranial atherosclerotic stenosis [10].

Lipoprotein(a) Lp(a) was evaluated in a Chinese cross-sectional study that included 75305 adults with Lp(a) measurements that underwent carotid ultrasound [11]. A multiple logistic regression analysis showed that participants with Lp(a) levels ≥ 50 mg/dL had an increased risk of carotid intima-media thickness (cIMT) ≥ 1.0 mm (OR = 1.138, 95% CI, 1.071–1.208) and carotid plaque (OR = 1.296, 95% CI, 1.219–1.377) compared with those with Lp(a) levels < 50 mg/dL [11].

Soluble lectin-like oxidized low-density lipoprotein receptor 1 (sLOX-1) is a soluble scavenger receptor released by protease hydrolysis of LOX-1 on the cell surface, and therefore, sLOX-1 levels reflect the expression level of LOX-1 [12]. sLOX-1 has been shown to be associated with the progression of atherosclerosis in endothelial cells and to be an independent predictor of functional outcome in patients with LAA IS [13]. It has also been shown to be associated with vulnerability of intracranial plaques and suggested to be useful as a supplement to high-resolution magnetic resonance vessel wall imaging to predict stroke recurrence [12]. A study by Bang et al. analyzed the association between serum lipid indices other than LDL-c and the occurrence of symptomatic cervicocephalic atherosclerosis in a retrospective study using data from a prospective registry [9]. This study included 1049 patients, divided between an LAA stroke group (*n* = 247) and a non-LAA group (*n* = 802). Patients with LAA stroke had higher triglyceride, a high-density lipoprotein cholesterol (HDL-c) ratio and non-HDL-c levels. After adjusting for age, risk factors, body mass index and premorbid statin use, the highest quartile of triglycerides was significantly associated with the occurrence of LAA stroke. These findings could explain why atherosclerosis in some patients progresses even after attaining intensive LDL-c reduction. Triglycerides have been reported to be a driving factor behind the progression of mild-to-moderate non-hemodynamically significant stenotic lesions [9].

Triglyceride levels were also studied by Jiang et al., who looked at insulin resistance and its marker triglyceride-glucose index [14]. This retrospective cross-sectional study looked at 2836 patients admitted to the Ma’anshan People’s Hospital due to acute IS, again divided between an LAA stroke group (*n* = 458) and a non-LAA stroke group (*n* = 2378). Age, hypertension and the triglyceride-glucose index were identified as predictors of LAA stroke, with a positive correlation between the triglyceride-glucose index and LAA stroke incidence, even after adjusting for related risk factors. While the retrospective design of the study prevents further conclusions, the authors hypothesize that the high triglyceride-glucose index might have contributed to the development of unstable atherosclerotic plaques by affecting inflammatory active substances, and subsequently led to clinical events [14].

Other parameters not directly involved in lipid pathways have been studied as possible biomarkers of LAA stroke. Cystatin C (CysC), a marker of renal function that is a risk factor for cardiovascular disease, was found to be independently associated with symptomatic extracranial internal carotid artery (ICA) stenosis, but not with intracranial ICA/middle cerebral artery stenosis in Japanese patients with noncardioembolic stroke [15].

Plasma levels of tumor-necrosis-factor-related apoptosis-inducing ligand (TRAIL) were reported to be significantly lower for LAA patients than controls (*p* < 0.001) [16]. In a Chinese study that included 132 LAA stroke patients and 60 control patients, the plasma TRAIL level was negatively correlated with the IS functional prognosis evaluated with the modified Rankin scale at 3 months (r = −0.372, *p* < 0.001). The optimal cut-off value of TRAIL for functional prognosis determination was 848.63 pg/mL. The sensitivity and specificity of this cut-off value were 63.1% and 86.2%, respectively [16].

Levels of adiponectin and endothelial progenitor cells (EPCs) in LAA stroke patients (*n* = 127) were found to be significantly lower compared with matched controls (*n* = 58) (*p* < 0.05) [17]. Both adiponectin and EPCs have been proposed to have anti-atherosclerosis effects [17].

### 3.2. Inflammatory Biomarkers

#### 3.2.1. Proteins

Inflammation is involved in the development, progression and rupture of atherosclerotic plaques. Several inflammatory substances have been studied as possible biomarkers of atherosclerosis and LAA stroke. C-reactive protein (CRP) and, in particular, high-sensitivity CRP (hs-CRP) have been associated with the presence of unstable carotid artery stenosis. Also, tumor necrosis factor α (TNF-α), an inflammatory cytokine, numerous cell adhesion molecules and matrix metalloproteinases were shown to have a positive correlation. The rationale for each is easily understood when considering the pathophysiological process of atherosclerosis [8].

More recent studies continue to attest the relation between LAA stroke and inflammation. A study by Wei and Quan, which consisted of a protein–protein interaction (PPI) study done with recourse to big data tools, found that the pertinent nodes of the PPI network for LAA included Nuclear Factor Kappa B Subunit 1 (NFκB1), interleukin-6 (IL-6), TNF-α and apolipoprotein B, which are right at the center of the inflammatory atherosclerotic cascade [18].

In a study including 75 Spanish patients with first-ever symptomatic intracranial stenosis that were prospectively followed, progression of symptomatic intracranial LAA was associated with a proinflammatory state [19]. Patients with intracranial LAA showed increased levels of CRP (CRP > 5.5 mg/L; HR, 5.4 [2.3 to 12.7]; *p* = 0.0001) and plasminogen activator inhibitor-1 (PAI-1) (PAI-1 > 23.1 ng/mL; HR, 2.4 [1.0 to 5.8]; *p* = 0.05) [19].

In a Swedish study that included 162 patients with cryptogenic stroke and 73 control patients and in which blood was collected in the acute phase and after 3 months, increased eotaxin and monocyte chemoattractant protein (MCP-1) were found to be the main markers suggestive of occult atherosclerosis [20]. Eotaxin is a selective chemoattractant for T2 lymphocytes, basophils and eosinophils that is released by activated endothelial cells.

A small study that included 58 patients with carotid stenosis showed that patients with a vulnerable plaque showed upregulation of the proinflammatory cytokines (IL-6 and TNFα), endothelial activation markers (E-selectin and vascular cell adhesion molecule 1) and inflammation markers (hs-CRP and pentraxin (PTX3)), and downregulation of the anti-inflammatory markers (adiponectin and IL-10) [21]. Vulnerable plaque was defined in this study by using ultrasonography and included low-echo plaques and mixed-echo plaques [21].

In recent years, IL-6 has been in the spotlight as a biomarker of atherosclerosis. It is a proinflammatory cytokine secreted by activated monocytes, macrophages, endothelial cells, adipocytes, fibroblasts and T-helper-2 cells, typically after stimulation by interleukin-1 or TNF-α. In a prospective population-based cohort study that included 4334 patients, and in which Duplex carotid ultrasound was performed at baseline and at 5 years of follow-up, log IL-6 predicted plaque severity (β = 0.09, *p* = 1.3 × 10^−3^), vulnerability (OR, 1.21 [95% CI, 1.05−1.40]; *p* = 7.4 × 10^−3^, *E*-value = 1.71) and progression (OR, 1.44 [95% CI, 1.23–1.69], *p* = 9.1 × 10^−6^, *E*-value 2.24) [22]. In participants with a >50% predicted probability of progression, mean log IL-6 was 0.54, corresponding to 2.0 pg/mL. The authors of this work hypothesized that the 2.0 pg/mL cut-off of IL-6 could facilitate the selection of individuals that would benefit from anti-IL-6 drugs for stroke prevention [22]. A systematic review by Papadopoulos et al., which reunited over 2500 patients, established higher levels of IL-6 as a risk factor for incident IS. Unfortunately, no stroke etiology analysis was presented [23].

The serum level of chitinase-3-like protein 1 (YKL-40) was also found to be a significant and independent biomarker for predicting the clinical outcome of LAA stroke [24]. YKL-40 is a glycoprotein produced by inflammatory, cancer and stem cells.

#### 3.2.2. White Blood Cells

Regarding blood cells count, a high ratio of monocytes to lymphocytes has been reported to be a predictor of poor functional prognosis in a study of 296 Chinese patients with LAA that had a development and a validation cohort [25]. The neutrophil-to-lymphocyte ratio—a biomarker that combines two faces of the immune system, the innate immune response, mainly due to neutrophils, and adaptive immunity, supported by lymphocytes—has been consistently associated with the presence and number of carotid plaques as well as with the prognosis of LAA patients with IS [26]. In a retrospective study that included 588 Chinese patients with acute IS and 309 healthy controls without carotid plaques, the admission neutrophil-to-lymphocyte ratio (NLR) was found to be an independent predictor of vulnerable carotid plaques after controlling for age, gender, diabetes mellitus and systolic blood pressure [27]. Corriere et al. also reported that NLR was a strong predictor of the presence and number of carotid atherosclerotic plaques. In this study that included 324 patients, >65 years old, NLR was a better predictor of the presence of carotid plaques than CRP and fibrinogen [28]. The cut-off point established in this study was NLR > 3.68. Corriere at al. considered that the mechanism by which NLR could be linked to atherosclerosis is probably based on activation of neutrophils within the plaque, leading to infiltration and progression of vessel wall lesions, underlined by inflammation and protein hydrolysis [28]. Another possible mechanism linking NLR to the development of carotid atherosclerotic plaques may be related to a dysfunction of the autonomic nervous system [28]. Neutrophils have adrenergic receptors, and the number and function of neutrophils are stimulated by sympathetic nerve endings. An increased sympathetic tone is also positively associated with increased oxygen consumption and production of proinflammatory cytokines, such as IL-6 and TNF-α [28,29]. An imbalance of autonomic nervous system may therefore be involved in the development and progression of atherosclerosis [28,30].

A retrospective Chinese study that included 487 patients with acute IS found that NLR and higher neutrophil counts were independently associated with the high stress hyperglycemia ratio in patients with LAA stroke [31]. SHR is used to assess stress hyperglycemia which is associated with the functional prognosis of IS [31]. The authors speculate that stress hyperglycemia may promote the progression of atherosclerosis by activating peripheral blood lymphocytes and neutrophils and disrupting the blood–brain barrier in IS patients with LAA [31].

Although NLR is a proven independent prognostic factor for morbidity and mortality in several diseases, its normal cut-off value is still under debate [26]. The Rotterdam study reported that male gender and older age (>85 years old) were associated with higher mean NLRs than in the general population, therefore these variables will need to be taken into account to establish an NLR cut-off point [32].

Furthermore, neutrophil extracellular traps (NETs) that exhibit proinflammatory and prothrombotic properties have been studied in patients with carotid artery stenosis. NETs are networks of extracellular fibers, primarily composed of DNA from neutrophils, that bind pathogens. In a study that included 39 consecutive Japanese patients that underwent carotid artery stenting with dual protection and in which local arterial blood was aspirated at the stent site to measure peptidylarginine deiminase 4 (PAD4), which is essential for the formation of NETs, in a multiple linear regression analysis, PAD4 was correlated with the NLR (*p* = 0.01) and plaque ulceration (*p* = 0.01, cut-off value: 0.49, odds ratio: 19.3) [33].

### 3.3. Metabolomics

A study that analyzed, by non-targeted metabolomics based on liquid chromatographymass spectrometry, 49 patients with LAA and 50 patients with SVO and 50 matched healthy controls, found differences in the metabolic profiling between the LAA and SVO groups. There were eight different metabolites, including L-pipecolic acid, 1-Methylhistidine, PE, LysoPE and LysoPC, which affected glycerophospholipid metabolism, glycosylphosphatidylinositol-anchor biosynthesis, histidine metabolism and lysine degradation [34].

### 3.4. RNA Biomarkers

#### 3.4.1. MicroRNAs

MicroRNAs (miRNA) are small, noncoding RNA particles, composed of 18 to 22 nucleotides that participate at the post-transcriptional level of gene expression, inhibiting translation or causing degradation of the messenger RNA. Multiple studies have documented their role in multiple biological processes and diseases, such as stroke and atherosclerosis. The Tampere vascular study assessed miRNA expression profiles in human atherosclerotic plaques and compared them with nonatherosclerotic arteries [3]. They found 10 miRNA with a statistically significant difference in expression, of which miR-21, miR-34a, miR-146b-5p and miR-210 had a higher expression in atherosclerotic arteries. The genes regulated by those miRNAs were involved in signal transduction, regulation of transcription and vesicular transport [3].

While there can be many possible miRNA biomarkers, they can also be more specific than conventional ones. A study by Xuan et al. assessed miR-137, a miRNA involved in the function of vascular endothelial and smooth muscle cells and in angiogenesis, as a possible biomarker for the diagnosis of cerebral atherosclerosis [35]. They compared miR-137 expression between 52 patients admitted with LAA IS and 46 controls. As expected, they found a downregulation of miR-137 expression in patients with LAA stroke, which was well correlated with other assessed biomarkers (total cholesterol, LDL-c and hs-CRP). However, when different biomarkers were compared, the area under the curve (AUC) for miR-137 was 0.908 (specificity of 87%), while the AUCs for total cholesterol, LDL-c and hs-CRP, were 0.810 (specificity of 71.7%), 0.819 (specificity of 84.8%) and 0.624 (specificity of 32.6%), respectively. They concluded that miR-137 had a high diagnostic value for LAA stroke etiology, with a specificity higher than other established biomarkers for atherosclerosis [35].

There are some research studies on the association between the expression of small noncoding microRNAs with carotid plaque development and vulnerability [36]. However, data are inconsistent. Also, all major studies regarding microRNAs and carotid atherosclerotic plaques were conducted either on cell cultures or animal models. There are very few studies that were conducted on humans. Therefore, the results of most of these studies cannot be automatically extrapolated to humans [4,36]. There is a lack of robust multicentric studies proving the reproducibility of miRNA biomarkers and a clear practical applicability for their use [3].

MiRNAs, however, have features that favor their use as biomarkers such as being highly stable in blood and having a good correlation between plasmatic and tissue levels, which makes them easily accessible and reliable. Also, new well-defined protocols have been developed for their detection, extraction and isolation. Finally, with the recent development of oligonucleotide/antisense therapies, miRNA remain a good hypothetical therapeutic target. Regulation of its expression could theoretically modulate disease development and effectively prevent the atherosclerotic process.

#### 3.4.2. Circular RNA

More recently, circular RNA (circRNA), a different type of RNA molecule, has been studied as an IS biomarker. These molecules are evolutionarily conserved and participate in regulatory functions, acting as sponges to sequester miRNA or RNA-binding proteins, that way modulating the expression of target genes. They are claimed to be more stable than other linear RNAs due to their resistance to endonuclease activity. A Spanish study profiled circRNAs in the peripheral blood of 30 patients admitted due to IS using a circRNA array and found different profiles when comparing atherosclerotic versus cardioembolic stroke [37]. The circRNA that were identified were predicted to interact with miRNAs involved in fatty acid biosynthesis and metabolism, lysine degradation, arrhythmogenic right ventricular cardiomyopathy, adrenergic signaling in cardyomyocytes and hypertrophic cardiomyopathy, some of which are processes related to atherosclerosis. However, the underlying mechanisms remain unclear, and this was a study with a small sample size [37].

Ostolaza et al. [37] found when comparing LAA and undetermined etiology stroke, that there were 226 circRNAs differentially expressed, 87 circRNAs upregulated and 139 circRNAs downregulated, of which only one circRNA expression was more than quadrupled. Differential expression of circRNAs in LAA stroke and cardiac embolism was verified by qRT-PCR. It was found that only ubiquitin Amur52 ribosomal protein fusion product 1 (UBA52) gene HSA_circRNA_102488, which originated on chromosome 19, had statistically significant changes between different etiological subtypes, and the RBP site of hsa_circRNA_102488 was clustered around the “Argonaute RISC Catalytic Component 2” and “Fused in Sarcoma” proteins. Additionally, functional analysis showed that differentially expressed circRNAs mainly interacted with stroke-related miRNAs [38].

#### 3.4.3. Transfer-RNA-Derived Small RNAs

Transfer-RNA-derived small RNAs (tsRNAs) are fragments that originate from mature or precursor tRNAs and are a subclass of sRNAs. TsRNAs exert extensive functions, including gene silencing, translational regulation and reverse transcriptional regulation, affecting cell viability and differentiation and participating in pathological processes of various diseases [39]. A study performed in a Chinese sample of patients found that tsRNAs targeting circulating exosomal tsRNAs could be potential biomarkers for diagnosing LAA stroke [39]. In this study, subjects were divided into a validation set (30 LAA:30 NC) and a replication set (120 LAA:105 NC:110 SAO). The study showed that exosomal tsRNAs were better at differentiating LAA stroke from other groups than plasma tsRNAs [39]. Namely, combined tRF-19-INVDRIFU and tRF-38-Q99P9P9NH57S36D1 had greater diagnostic efficacy. Furthermore, exo-tRF-19-INVDRIFU contributed to assess plaque rupture risk [39].

#### 3.4.4. Long Noncoding RNA

A study that analyzed patients with advanced carotid artery atherosclerotic lesions from the Biobank of Karolinska Endarterectomies profiled differences in the RNA transcript. The long noncoding RNA myocardial infarction-associated transcript (lncRNA MIAT) was identified as the most upregulated noncoding RNA transcript in carotid plaques compared with nonatherosclerotic control arteries; this was confirmed by quantitative real-time polymerase chain reaction and in situ hybridization [40]. MIAT may play a role in regulating proliferation and transdifferentiation of arterial smooth muscle cell, inflammatory activity and macrophages, as well as during atherosclerotic plaque development and progression [40].Targeting MIAT could serve as a novel molecular treatment strategy to limit vascular inflammation and atherosclerosis progression [40].

Some studies have evaluated the use of LncRNA to differentiate stroke etiologies. In a prospective observational study that included 80 Chinese patients with acute IS stroke (40 with LAA and 40 with cardioembolism), patients with cardioembolism had considerably higher plasma levels of long noncoding mitochondrially encoded long noncoding (lncRNA) cardiac-associated *RNA* (LIPCAR) than patients with LAA [41].

Also, Zinc finger antisense 1 (ZFAS1), a newly identified lncRNA, was shown in LAA IS to be significantly downregulated when compared to non-LAA IS and controls [3].

These findings suggest that the levels of lncRNA LIPCAR and ZFAS1 may play a potential role in the distinction between LAA and other subtypes in patients with acute IS [41,42].

### 3.5. Genetic Biomarkers

Genetics hold many answers to pivotal questions related to the atherosclerotic disease processes, even if we still do not have the right tools to understand them. This same premise also applies to biomarkers related to LAA stroke, with genetic polymorphisms being assessed as potential ways to solve clinical problems.

Regarding atherosclerotic IS, cystatin C (CysC) is known to be involved in atherosclerotic plaque remodeling, being independently associated with cerebral artery stenosis and prognosis in stroke patients. With that in mind, two selected single-nucleotide polymorphisms of the *CST3* gene were evaluated in 3833 subjects as possible biomarkers of LAA stroke [43]. This was a multicenter, prospective registry study that only included Chinese patients. No statistically significant association was found between any allele and the occurrence of LAA stroke, however, carriers of the T alelle of SNP rs13038505 tended to have a lower proportion of LAA stroke. The specific functions of the two SNPs and their causal relationship with CysC concentration remains to be clarified [43].

Another study looked at methylation alteration patterns in candidate genes in LAA stroke, as they are reported to be associated with the development of IS [44]. This study was also done in Chinese patients and evaluated 301 patients with LAA stroke, with age- and sex-matched controls. A total of 1012 annotated CpG loci in 672 genes, which were involved in different aspects of the nervous system, were identified as differentially methylated based on the established threshold. The investigators looked particularly at the gene *MTRNR2L8*, that functions as a neuroprotective factor, and its promotor methylation status, and acknowledged its high predictive value in the prognosis of IS, possibly serving as a guide to its prevention and clinical diagnosis [44].

In a genetic association study, interleukin IL-6, IL-1β, monocyte chemoattractant protein-1 macrophage inflammatory protein-1α, E-selectin, intercellular adhesion molecule 1, matrix metalloproteinase-3 and nine gene variants were found to be independently and significantly associated with atherosclerotic internal carotid stenosis [45].

### 3.6. Biomarkers Found in Retrieved Thrombi

The advent of mechanical thrombectomy made possible the histologic analysis of retrieved thrombi to assess their origin. This was done by Wang et al., who studied thrombi in a prospective multicenter cross-sectional study of patients with IS who had undergone thrombectomy [46]. Even though it was a small study with a total amount of 94 patients, of which 56 had a cardioembolic etiology and 36 an atherothrombotic cause, they found that thrombi with an LAA source had significantly higher actin and CD105 levels than thrombi with a cardioembolic source. The authors hypothesized that actin levels could predict an LAA source because of its association with plaque formation and stability [46]. An increased finding of a localized pattern of the oxidative stress marker 4-hydroxyl-2-nonenal was reported in clots retrieved from patients with LAA when compared to patients with cardioembolic and cryptogenic stroke (*p* < 0.01), suggesting that it may be a new marker of LAA [47].

## 4. Conclusions

Most blood biomarkers associated with LAA stroke evaluated LAA diagnosis, patients’ prognosis, carotid atheroma diagnosis and the presence of vulnerable plaques (Figure 1).

In the future, biomarkers could be used to stratify patients according to disease severity and to evaluate their response to treatment. Overall, there is a preponderance of retrospective studies evaluating biomarkers associated with LAA stroke. Most studies evaluated only Chinese patients and were not externally validated. External validation is needed to evaluate the reproducibility and generalizability of the findings of these studies in different geographical locations and settings. Also, only a small fraction of studies used a development and an internal validation cohort. RNA biomarkers still need further studies to assess their clinical utility.

To adopt these biomarkers in clinical practice, we will need robust multicentric studies proving their reproducibility and a clear practical applicability for their use.

## Figures and Tables

**Figure 1 biomolecules-13-01251-f001:**
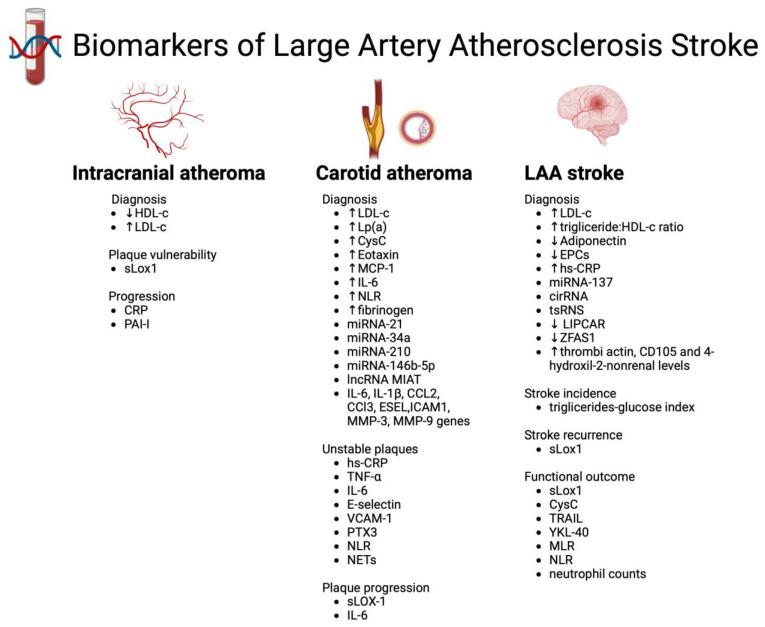
Summary of the main biomarkers associated with LAA stroke, ↑—increased levels, ↓ decreased levels.

## Data Availability

No new data were created.

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
