# Peer review of "Update on Biomarkers Associated with Large-Artery Atherosclerosis Stroke"

_biomolecules, 2023, doi:10.3390/biom13081251_

Round 1

Reviewer 1 Report

Long sentences are present through out the manuscript. For instance: page 1 line 36-42. Please try to rephrase them. Page 1, line 17: please rephrase your sentence. Page 1, line 93: sLOX-1 abbreviation does not need to be reminded. Page 4, line 186: monocytes to lymphocytes ratio does need to be abbreviated. Page 4, line 189: neutrophil-to-lymphocite ratio needs to be abbreviated as it repeats later. Page 5, line 237: please rephrase your sentence. Page 6, line 284: there is no need for abbreviations.

none

Author Response

We have rephrased the sentence present in page 1, line 36-42:

“As the name implies, atherosclerosis is the underlying pathophysiology in LAA stroke. Atheroma is typically located at the bifurcation of arteries, where turbulent flow is highest. Several mechanisms contribute to the formation and development of the atheromatous lesion. Initially, induced endothelial dysfunction, potentiated by inflammation and hypercholesterolemia, lead to increased permeability, with entrance of oxidated low-density lipoproteins in the subendothelial space of the intima. Meanwhile, at the endothelial surface, the expression of adhesion molecules initiates platelet aggregation and lymphocyte/monocyte adhesion and infiltration. In the intima, these monocytes mature into macrophages, which take up oxidized low-density lipoprotein and transform into foam cell. As this occurs, vascular smooth muscle cells shift from a contractile phenotype to an active synthetic phenotype, producing extracellular matrix and thereby gradually transforming the lesion into a fibrous plaque [2].”

Page 1, line 17:

“Some biomarkers were associated with the prognosis of LAA stroke. Namely, increased levels of IL-6 were associated with risk of progression of the atherosclerotic disease.”

Page 1, line 93

We eliminated the reminder of the sLOX-1 abbreviation.

Page 4, line 186

We eliminated the monocytes to lymphocytes ratio abbreviation.

Page 4, line 189

We abbreviated the neutrophil-to-lymphocyte ratio.

Page 5, line 237: please rephrase your sentence.

We have rephrased this paragraph. It is now stated “There are some research studies on the association between the expression of small non-coding microRNAs with carotid plaque development and vulnerability [36]. However, data is inconsistent. Also, all major studies regarding microRNAs and carotid atherosclerotic plaques were conducted either on cell cultures or animal models.  There are very few studies that were conducted on humans. Therefore, the results of most of these studies cannot be automatically extrapolated to humans [4,36]. There is a lack of robust multicentric studies proving the reproducibility of miRNA biomarkers and a clear practical applicability for their use [3].”

Page 6, line 284: there is no need for abbreviations

We erased the abbreviations.

Reviewer 2 Report

The authors performed a narrative review of the literature published in recent years regarding biomarkers associated with LAA stroke in humans.

The discussion of the available data is quite scarce. I would have appreciated a more detailed revision of the literature. It is not sufficient to state that external validation is needed.

Although the list of possible biomarkers is quite extensive, I think that something else should be added, such as lncRNA: are they involved in LAA?

The English is easy to read, but there are some mistakes that should be corrected:

1)     Line 17: Increased levels of IL-6 and have been…. Something is missing

2)     Line 157: incrased…..I think they meant increased

3)     Line 237: There is a lack of are robust……. Something is wrong

Author Response

The authors performed a narrative review of the literature published in recent years regarding biomarkers associated with LAA stroke in humans.

The discussion of the available data is quite scarce. I would have appreciated a more detailed revision of the literature. It is not sufficient to state that external validation is needed.

We included more data and references regarding inflammatory biomarkers and RNA. To clarify, our previous statement regarding the need for external validation, we added to the conclusions section the reasoning for which external validation is needed. It is now stated in the conclusion section: Most studies evaluated only Chinese patients and were not externally validated. External validation is needed to evaluate the reproducibility and generalizability of the findings of these studies namely in different geographical locations and settings. Also, only a small fraction of studies used a development and an internal validation cohort.”

Although the list of possible biomarkers is quite extensive, I think that something else should be added, such as lncRNA: are they involved in LAA?

We added some information regarding LncRNA.

It is now stated:

3.4.4 Long non-coding RNA

A study that analyzed patients with advanced carotid artery atherosclerotic lesions from the Biobank of Karolinska Endarterectomies profiled differences in RNA transcript. The long noncoding RNA Myocardial Infarction Associated Transcriot (lncRNA MIAT) was identified as the most upregulated noncoding RNA transcript in carotid plaques compared with nonatherosclerotic control arteries, which was confirmed by quantitative real-time polymerase chain reaction and in situ hybridization [41]. MIAT may play a role in regulating proliferation and transdifferentiation of arterial smooth muscle cell and inflammatory activity and macrophages, as well as during atherosclerotic plaque development and progression [41]. Targeting MIAT could serve as a novel molecular treatment strategy to limit vascular inflammation and atherosclerosis progression [41].

Some studies have evaluated the use of LncRNA to differentiate stroke etiologies. In a prospective observational study that included 80 Chinese patients with acute IS stroke (40 with LAA and 40 with cardioembolism), patients with cardioembolism had considerably higher plasma levels of long non-coding Mitochondrially Encoded Long Non-Coding (lncRNA) Cardiac Associated RNA (LIPCAR) than patients with LAA [42].

Also, Zinc finger antisense 1 (ZFAS1), a newly identified lncRNA was shown in LAA IS to be significantly downregulated when compared to non-LAA IS and controls [3]

These findings suggest that the levels of lncRNA LIPCAR and ZFAS1 may play a potential role in the distinction between LAA and other subtypes in patients with acute IS [42,43].”

The English is easy to read, but there are some mistakes that should be corrected:

1)     Line 17: Increased levels of IL-6 and have been…. Something is missing

2)     Line 157: incrased…..I think they meant increased

3)     Line 237: There is a lack of are robust……. Something is wrong

Thank you; all the previous points were amended in the text. 

Author Response

This review by Rosario and Fonseca focused on emerging biomarkers associated with Large Artery Atherosclerosis Stroke. Unfortunately, in this some background references were missed and not argued. For example, the role of some circulating miRNA-195-5p and -451a (Giordano M. et al., J Clin Med 2019; Giordano M. et al., Int J Mol Sci 2020; Giordano M. et al., Life 2022), of some circulating  long non coding RNAs (Z.Z. Li uet al., Neurol Ther 2023), as well as of some circulating circular RNAs (X. Wang et al., Ann Clin Trnsl Neurol 2023) and exosomal tsRNAs (K. Yang et al., Clin Transl Med 2023). Moreover, the poptential implication of a derangement of the interplay between innate and adaptive immunity (A. Buonacera et al., Int J Mol Sci 2022) should be also brought to the fore, as recently emphasized by X. Feng et al., (Front Endocrinol 2023), showing the role of biomarker of neutrophil to lymphocyte ratio (NLR) and neutrophil count.

As a matter of fact, some years ago NLR was shown in a pioneer study to be associated with the presence of atheroislerotic plaques (T. Corriere et al., Nutr Metab Cardiovasc Dis 2018), so providing by an easily available biomarker further evidence in favour of a potential involvement of an immunologic dysfunnction in the pathogenetic chain of large arteries ischemic stroke.

Authors are therefore kindly asked to expand the reference list and to discuss in more details some pathogenetic aspects linking emerging biomarkers to  Large Artery Atherosclerosis Stroke.  

Thank you for your comment. We included most of the references that were mentioned by the reviewer in our manuscript. We have therefore extensively edited the sections regarding inflammatory biomarkers and RNA. We read the articles of 2019 and 2020 by Giordano, however these articles do not concern LAA stroke. These articles evaluated the use of biomarkers to diagnose of TIA or ischemic stroke in general. There is no mention in those articles of a specific etiology of the TIA/strokes (cardioembolic, small vessels, undetermined, LAA, other specific etiology). Also, Giordano´s article in Life 2022 is about hemorrhagic stroke which is an anatomopathological different type of stroke from the one that is covered in this review. Therefore, none of these articles could be included in our review that refers specifically to LAA ischemic stroke.

It now stated in section 3.2:

Corriere et al. also reported that NLR was a strong predictor of the presence and number of carotid atherosclerotic plaques.In this study that included 324 patients, >65 years-old, NLR was a better predictor of the presence of carotid plaques than CRP and fibrinogen [28]. The cut-off point established in this study was NLR > 3.68. Corriere at al. considered that the mechanism by which NLR could be linked to atherosclerosis is probably based on activation of neutrophils within the plaque, leading to infiltration and progression of vessel wall lesions, underlined by inflammation and protein hydrolysis [28]. Another possible mechanism linking NLR to development of carotid atherosclerotic plaques may be related to a dysfunction of the autonomic nervous system [28]. Neutrophils have adrenergic receptors, and the number and function of neutrophils are stimulated by sympathetic nerve endings. An increased sympathetic tone is also positively associated with increased oxygen consumption and production of proinflammatory cytokines, such as IL-6 and TNF- α [28, 29]. An imbalance of autonomic nervous system may therefore be involved in the development and progression of atherosclerosis [28,30].

A retrospective Chinese study that included 487 patients with acute IS found that NLR and higher neutrophil counts were independently associated with high stress hyperglycemia ratio in patients with LAA stroke [31]. SHR is used to assess stress hyperglycemia which is associated with functional prognosis of IS [31]. The authors speculate that stress hyperglycemia may promote the progression of atherosclerosis by activating peripheral blood lymphocytes and neutrophils and disrupting the blood brain barrier in IS patients with LAA [31].

Although NLR is a proven independent prognostic factor for morbidity and mortality in several diseases, its normal cut-off value is still under debate [26]. The Rotterdam study, reported that male gender and older age (>85 years-old) were associated with higher mean NLRs in the general population, therefore these variables will need to be taken into account to establish a NLR cut-off point [32].

An in section 3.4:

Ostolaza al. [37] found when comparing LAA and undetermined etiology stroke, that there were 226 circRNAs differentially expressed, 87 circRNAs upregulated, and 139 circRNAs downregulated, of which only one circRNA expression was more than quadrupled. Differential expression of circRNAs in LAA stroke and cardiac embolism was verified by qRT-PCR. It was found that only ubiquitin Amur52 ribosomal protein fusion product 1 (UBA52) gene HSA_circRNA_102488, which originated on chromosome 19, had statistically significant changes between different etiological subtypes, and the RBP site of hsa_circRNA_102488 was clustered around the “Argonaute RISC Catalytic Component 2” and “Fused in Sarcoma” proteins. Additionally, functional analysis showed that differentially expressed circRNAs mainly interacted with stroke-related miRNAs [38].

3.4.3 Transfer RNA-derived small RNAs

Transfer RNA-derived small RNAs (tsRNAs) are fragments that originate from mature or precursor tRNAs and are a subclass of sRNAs. TsRNAs exert extensive functions, including gene silencing, translational regulation, and reverse transcriptional regulation, affecting cell viability and differentiation and participating in pathological processes of various diseases [39]. A study performed in a Chinese sample of patients found that tsRNAs targeting circulating exosomal tsRNAs could be potential biomarkers for diagnosing LAA stroke [40]. In this study, subjects were divided into a validation set (30 LAA: 30 NC) and a replication set (120 LAA: 105 NC: 110 SAO). The study showed that , exosomal tsRNAs were better in differentiating LAA stroke from other groups than plasma tsRNAs [40]. Namely, combined tRF-19-INVDRIFU and tRF-38-Q99P9P9NH57S36D1 had greater diagnostic efficacy. Furthermore, exo-tRF-19-INVDRIFU contributed to assess plaque rupture risk [40].

3.4.4 Long non-coding RNA

A study that analyzed patients with advanced carotid artery atherosclerotic lesions from the Biobank of Karolinska Endarterectomies profiled differences in RNA transcript. The long noncoding RNA Myocardial Infarction Associated TRanscriot (lncRNA MIAT) was identified as the most upregulated noncoding RNA transcript in carotid plaques compared with nonatherosclerotic control arteries, which was confirmed by quantitative real-time polymerase chain reaction and in situ hybridization [41]. MIAT may play a role in regulating proliferation and transdifferentiation of arterial smooth muscle cell and inflammatory activity and macrophages, as well as during atherosclerotic plaque development and progression [41].Targeting MIAT could serve as a novel molecular treatment strategy to limit vascular inflammation and atherosclerosis progression [41].

Some studies have evaluated the use of LncRNA to differentiate stroke etiologies.In a prospective observational study that included 80 Chinese patients with acute IS stroke (40 with LAA and 40 with cardioembolism), patients with cardioembolism had considerably higher plasma levels of long non-coding Mitochondrially Encoded Long Non-Coding (lncRNA) Cardiac Associated RNA (LIPCAR) than patients with LAA [42].

Also, Zinc finger antisense 1 (ZFAS1), a newly identified lncRNA was shown in LAA IS to be significantly downregulated when compared to non-LAA IS and controls [3]

These findings suggest that the levels of lncRNA LIPCAR and ZFAS1 may play a potential role in the distinction between LAA and other subtypes in patients with acute IS [42,43].

Reviewer 4 Report

Manuscript is a narrative review of the literature mainly published in the last five years with the aim of identifying biomarkers associated either with intracranial or extracranial LAA in humans.  It summarize chemical, inflammatory, metabolomics, genetic and  RNA biomarkers as well as biomarkers found in retrieved thrombi.

The manuscript is concise and clearly written, but it would greatly benefit from an illustrative picture or an overview table of biomarkers.

English is generally good, only minor corrections are needed.

Author Response

We added an illustrative picture (Figure 1).

Round 2

Reviewer 2 Report

The authors replied to all my comments and the manuscript is now acceptable for publication.

Reviewer 3 Report

No further concern.